# A Scoping Review of Transgender Policies in the 15 Most Commonly Played UK Professional Sports

**DOI:** 10.3390/ijerph20043568

**Published:** 2023-02-17

**Authors:** Michael McLarnon, Jane Thornton, Gail Knudson, Nigel Jones, Danny Glover, Andrew Murray, Michael Cummings, Neil Heron

**Affiliations:** 1Centre for Public Health, Queen’s University Belfast, Belfast BT12 6BA, UK; 2Schulich School of Medicine and Dentistry, Western University, London, ON N6A 3K7, Canada; 3Faculty of Medicine, University of British Columbia (UBC), Vancouver, BC V6T 1Z4, Canada; 4Medical Department, British Cycling, Manchester M11 4DQ, UK; 5Medical and Scientific Department, Ladies European Tour (Various), Denham UB9 5PG, UK; 6Sport and Exercise, University of Edinburgh, Edinburgh EH8 9YL, UK; 7School of Medicine, Keele University, Staffordshire ST5 5BG, UK

**Keywords:** scoping review, transgender and gender-diverse (TGD), TGD athletes, transitioning, guidelines, policies, fairness, inclusion, safety, testosterone, 2021 IOC Framework on Fairness, inclusion and non-discrimination

## Abstract

Introduction: There has been much debate recently on the participation of transgender and gender-diverse (TGD) athletes in sport, particularly in relation to fairness, safety and inclusion. The 2021 IOC Framework on Fairness, Inclusion and Non-discrimination acknowledges the central role that eligibility criteria play in ensuring fairness, particularly in the female category, and states that athletes should not be excluded solely on the basis of their TGD identity. Aims: To identify policies that address TGD athlete participation in the 15 major United Kingdom (UK) sporting organisations and to summarise the evidence for each of these policies. Methods: A scoping review of TGD policies from the 15 major UK sporting organisations. Results: Eleven of the governing bodies had publicly available TGD policies. Most of the sporting associations drew guidance from the official 2015 IOC Consensus Meeting on Sex Reassignment and Hyperandrogenism, particularly with regard to physiological testosterone levels. Many organisations referenced their policies as a guide for decision making but stated that they ultimately made case-by-case decisions on an athlete’s eligibility. Relevant considerations not addressed in most policies included pre- versus post-pubertal athletes, justification for testosterone thresholds, the length of time out of competitive action (if any) for transitioning athletes, the irreversible advantage from male puberty (if any), the responsibility for and frequency of follow up for hormonal testing and the consequences for athletes outside set testosterone limits. Conclusions: There is a lack of consensus among the top 15 UK sporting organizations relating to elite sport participation for TGD athletes. It would be useful for sport organizations to work together to develop greater standardization/consensus for TGD athlete policies, taking into consideration fairness, safety and inclusion in each sport.

## 1. Introduction

Fair, inclusive and non-discriminatory sport participation for all athletes, including transgender and gender-diverse (TGD) athletes, is currently a very topical issue within sport medicine and society in general. In 2021, the United Kingdom (UK) sport councils produced a document reviewing transgender inclusion in domestic sport (Guidance for Transgender Inclusion in Domestic Sport) [1]. This document concluded that there is no single solution to the issue, instead offering guiding principles to promote fairness, safety and inclusivity, but it recognised the presence of retained male advantage and noted that a decision should be made by sporting federations between fairness and inclusivity. The International Olympic Committee (IOC) recently released a framework for fairness, inclusion and non-discrimination on the basis of gender identity, putting greater emphasis on inclusion, although it deferred practical guidance and decision making to individual sports based on their specific circumstances [2]. Although some regulation has existed since the 1940s [3,4,5,6], consideration of TGD athlete participation and scientific research in the area were limited until recently. This review focused on policies related to transgender athletes and did not consider policies related to intersex athletes or those with differences in sexual differentiation (DSD).

Current sporting guidelines focus predominantly on testosterone-based regulations for trans women athletes participating in women’s sporting events. Researchers and policymakers cite variables that differentiate biological male and female athletic performance [7,8,9,10] produced by exposure to testosterone from the onset of male puberty, amongst other influences [5,11,12], which include height, muscle mass, pelvic and lower limb anatomy (e.g., the “Q” angle), reaction times, VO_2max_, strength and the mechanics of movements such as running [13,14,15,16,17,18,19,20,21,22,23]. Observational data relating to national and world records show a record gap between the sexes of 8–12% [7,24,25,26,27,28].

Sport participation for all needs to be promoted, as sport provides a plethora of benefits for both mental and physical health [29,30]. Indeed, TGD athletes must be able to participate in and have access to sport, and barriers to them doing so, in whichever category is determined most appropriate for the sport, should be addressed [2]. The barriers to subsequent participation by athletes in the female category, including TGD athletes, due to fairness, safety or other issues also need to be considered as part of this discussion [2,5,12].

### Aims

To understand how sporting organisations are navigating these issues, we aimed to conduct a scoping review of current guidelines from the 15 major UK sporting organisations with the highest participation numbers in order to:Determine those that have an existing TGD athlete policy;Review their specific guidance on eligibility (including testosterone-based regulations);Summarise the policies in different sports;Map the evidence gaps in the TGD policies across the different organisations.

## 2. Methods

A scoping review was considered the most appropriate methodological approach to address the research question. A scoping review is defined as an exploratory project that systematically maps the literature available on a topic, identifying key concepts, theories, sources of evidence and gaps in the literature [31,32]. Compared to systemic reviews, scoping reviews make it possible to investigate a less specific research question and work to inform future research [33]. This scoping review was based on the six-step methodological framework outlined by Arksey and O’Malley and informed by Levac et al. [34,35]. The Preferred Reporting Items for Systemic Review and Meta-Analysis (PRISMA) protocol was followed and the PRISMA extension for scoping reviews (PRISMA-ScR) checklist completed [31]. Throughout this review, we define transgender athletes as per the IOC; that is, “Someone who identifies with a gender that is different from the sex that they were assigned at birth. A transgender man is a man who was assigned female at birth; a transgender woman is a woman who was assigned male at birth” [2]. This review exclusively focuses on transgender athletes, without consideration for intersex athletes or athletes with differences in sex development (DSD) or related disorders.

### 2.1. Stage 1: Identifying the Research Question(s)

The following research objective was addressed in this report: to identify policies that address TGD athlete participation in the 15 major United Kingdom (UK) sporting organisations and to summarise the evidence for each of these policies.

To be included in this review, the sports had to be widely participated in at an amateur level within the UK. Participation levels were taken from the Sport England website via the Active Lives survey, as well as from independent reports from Harris Interactive, Statista and the Ipsos MORI sports tracker [36,37,38,39,40]. Polls from unofficial sources, such as major sporting magazines or online fan websites, were also reviewed, as were the Wikipedia pages for “Sport in England” and “Sport in the United Kingdom”.

### 2.2. Stage 2: Identifying Relevant Policies

Searches for TGD policies among the 15 major UK sports organisations were carried out from inception to December 2021 to find relevant guidelines to include in this review. This was undertaken by first identifying the sports with the highest levels of participation and then identifying the official governing bodies for each sport (both UK-wide and nationally) through simple internet searches. Following this, each organisation’s official website was searched for a transgender policy, and searches were also undertaken using search engines (Google search, Google Scholar). Where a policy was not identified or publicly available, direct inquiry with the organisation was attempted.

### 2.3. Stage 3: Policy Selection

TGD guidelines were accessed either via the official organisation websites or through a request addressed to a relevant representative. When separate organisations were present for the same sport in England, Scotland, Northern Ireland and Wales (for example, rugby union), the England guidelines were reviewed, as England represented the largest population of amateur athletes that would typically use such guidelines. Searches were carried out by MMcL and checked by NH, with AM reviewing any disagreements.

### 2.4. Stages 4 and 5: Charting the Data and Collating, Summarizing and Reporting Results

The following data were extracted for each sport: the primary governing organisation for that sport; the sports played within the remit of the organisation; the official TGD policy of the association; testosterone cut-off levels for transitioning athletes and their rationale, taking into consideration the different demands of the sports; the length of any required time out of competition during/post-gender-affirming therapy; the hormone eligibility evidence required by the governing body for sports participation; regulations around child or adolescent TGD participation; and specific case-by-case guidance for individual TGD athletes. Information was recorded using Microsoft Excel.

## 3. Results

### Sporting Organisation Guidelines

Data analysed for each sport, relevant UK sporting organisations and sports played within the organisations are summarised in Table 1. Although individual breakdowns of gender participation are not provided by the organisations, we know that there is generally a consistently higher proportion of male athletes engaging in amateur sport [37,41]. Data taken from an analysis of the Taking Part survey provide the proportions of male/female participants in popular sports in the UK [41] and are also included within Table 1. This information was available for 8/15 of the included sports. Where available, information on the specific guidance and rules that each organisation based their policy on is included.

Table 2 summarises the contents of the sport-specific guidelines for each organisation. As many policies are adopted from the IOC’s 2021 (and, before that, 2015) position [2], this has also been included for reference.

Twelve of the fifteen governing bodies had an official TGD policy (even if that policy was just to state that there were no interventions required for participation), of which eleven were publicly available (with the 2011 Rugby League guideline being referenced in an alternative publication but not available despite attempts to contact the governing body directly). Of these 12, all had pre-requisites for trans women athletes (testosterone monitoring, written medical documentation or clearance before a panel (Table 2)). Five of the twelve had separate guidance for adolescent/pre-pubescent versus post-pubertal athletic participation. Seven drew guidance from the official IOC Statement on Transgender Guidelines [44], particularly in reference to physiological testosterone levels. Many (8/12) organisations included the disclaimer that they made “case-by-case” decisions on an athlete’s competitive eligibility, using their policies only as a guide for decision making. The official 2021 IOC Framework also states that all athletes are approved on a case-by-case basis in their guidance, with wordings being very similar to this policy across each sport. The additional criteria assessed in case-by-case decisions are not discussed in any policy (with the exception of badminton, for which the guidance states that the duration of hormonal therapy/transitioning would be assessed individually to ascertain adequacy). Several policies have less strict restrictions on participation at the recreational/amateur level, including athletics, tennis and cycling. For athletics, the governing body, United Kingdom Athletics (UKA), specifies that restrictions do not apply to training, friendly/fun runs or mass participation events, such as charity events; restrictions are only applied to competitions where qualifying marks, selection standards and records are set. British Cycling stipulates that there are no regulations for recreational activities, and the Lawn Tennis Association states that recreational and friendly matches are exempt. Other policies do not specifically comment on training, recreational activity within clubs or friendly matches; therefore, their stances are unclear. Only two policies (Rugby Football Union and British Rowing) mention the specific involvement of or consultation with a medical professional/medical officer in their TGD policy development. Three of the policies are currently under review.

In two of the three sports without TGD policies, it could be proposed that physical differences in competitors are unlikely to result in harm to individuals or fellow competitors or involve a significant physical competitive advantage for males (“sex-affected sports”). These sports are snooker/billiards and motorsports. The third, however, is boxing; the British Boxing Board of Control has no official TGD participation policy. Although the Rugby Football League (RFL) refers to a TGD policy published in 2011, it is not publicly available and was not made available when we emailed the RFL contact on their website.

Notably, where policies exist, a definition of transgender is often provided to outline the remit of the policy. Policies relate to transgender as opposed to intersex athletes or athletes with DSD.

## 4. Discussion

This is the first scoping review to summarise the TGD policies of the 15 most widely played sports in the UK, detailing the individual protocols of each sporting governing body for TGD athletes. Our review showed that 80% (12/15) of the organisations have an official TGD athlete participation policy. The policies are highly heterogeneous, with different requirements and time frames for TGD athletes to be included in a class other than that of their biological sex. The majority of policies rely heavily on the official IOC Statement on Transgender policies [2,44], stating they will refer to it when making decisions on individual athletes and generally adopting a case-by-case approach when assessing TGD athletes. Only 38% (5/13) of the guidelines have separate guidance for adolescent or child athletes. There appears to be no difference in the rigour of assessment between non-contact and contact sports and, indeed, boxing has no current TGD policy. Criteria were globally much stricter for trans female athletes wishing to participate in the female category versus trans male athletes wishing to compete in the male category. Finally, although serum testosterone levels were specified for four sports (<2.5 nmol/L for cycling, <5 nmol/L for rugby union and rowing, and <10 nmol/L for badminton), for participation in the female events, few details are provided on what sporting organisation is responsible for the testing, the parameters around the testing and the steps taken if the athlete’s testosterone level is outside these stated ranges. Three sports (athletics, cycling and tennis) have more relaxed TGD policies for sport participation at the recreational level.

### 4.1. Adolescent and Child Participation

Five sports (cricket, rowing, cycling, golf and tennis) had pre-puberty guidelines for TGD sport participation, with most sports stating that puberty occurred from 16 years of age onwards. However, some children, including those under 16 years of age, may identify as TGD [45], and thus sport organisations may wish to formally address this within their TGD policies. The Guidelines for Creating Policies for Transgender Children in Recreational Sports, the national policy of the United States (US), assert that no hormonally based advantage exists for sexes prior to puberty [46], although other authors contend that there is pre-pubertal difference, in keeping with the concept of “mini-puberty”, as well as genes exclusively related to the Y-chromosome [47,48,49,50]. Furthermore, the onset of puberty shows some variance among individuals, as well as the sexes, and, therefore, it is difficult to set an arbitrary age limit via which athletes could be considered pre- and post-pubertal. Puberty is, however, rare before the ages of nine and eight in males and females, respectively [51,52,53]. There is, therefore, a need for TGD policies in sport to address pre- and post-pubertal athletic participation.

### 4.2. Anthropometric Sex Differences and Impact on Sport Performance

Although physical sex differences are noted (some in athletic and some in non-athletic populations), and certain sports seem to preferentially select at the extremes of the distribution for a trait (e.g., height in basketball), it is plausible that an anthropometric advantage could exist for TGD athletes in their sport. TGD policies should investigate this further within their cohort of athletes to determine whether it is the case. Similarly, when an athlete has increased muscle mass or body size relative to peers in their gender (outside of what can be observed within the given sex), they have greater potential to inflict injuries on others, potentially compromising safety [12].

Moreover, it has been reported that muscle nuclei that have been developed during periods of testosterone therapy remain in the muscle even after drug cessation and potentially confer a lifelong muscle-building advantage [54,55,56]. Therefore, although it is difficult to measure the magnitude of the advantage that might be obtained in a given sport for a range of physical metrics that differ between the sexes, these performance advantages are present partly due to androgenic male puberty and are not dependent on current circulating testosterone. Thus, TGD policies should consider whether the athlete has undergone puberty and what affect this could have on their participation in their chosen sport, as well as considering circulating testosterone levels.

### 4.3. Testosterone Cut-off Levels and Duration of Exclusion from Sporting Participation

#### 4.3.1. Transgender (Trans Women/Trans Female) and Gender-Diverse Athletes with Suppressed Testosterone

Our research indicated that, when specified, testosterone levels lower than 2.5 nmol/L for 24 months (cycling—UCI rules) and lower than 5 nmol/L for at least 12 months (rugby and rowing) are considered general cut-offs among UK sport organisation for trans women’s participation in female sport, although badminton guidelines specify a testosterone value lower than 10 nmol/l, in keeping with the 2015 IOC guidance [44]. The UCI has recently reduced its testosterone cut-off level to 2.5 nmol/L for two years [42], and the UK national governing body policy is likely to follow. However, adult males generally have testosterone levels at least 15 times higher than females of a similar age [57], and the 95% confidence interval for circulating testosterone in healthy premenopausal adult females is 0–1.7 nmol/L [57]. Thus, sport organisation need to consider what testosterone levels will apply to their sport and what lengths of time out of competition will apply. These findings may be particularly relevant for strength-based (requiring running and jumping) and contact sports; for example, rugby, football, boxing and martial arts. Different considerations may be needed for sports where physiological differences or potential safety concerns are not an issue and inclusivity can be prioritized.

Another important issue to consider for competitive, strength-based sport participation is non-compliance with the proposed upper limit for serum testosterone levels as measured by liquid chromatography–mass spectrometry. Only the IOC provides guidance for non-compliance with these limits [44], enforcing 12 months of exclusion following a supraphysiological result. However, little detail is provided on how this testing will be undertaken and whose responsibility it is to undertake it. The World Anti-Doping Association (WADA) has produced a physician guide for TUE applications for transgender athletes (accessed at https://www.wada-ama.org/en/resources/therapeutic-use-exemption/tue-physician-guidelines-transgender-athletes#resource-download, accessed on 10 January 2023) and suggests testosterone testing at least 1–2 times per year by the treating physician.

#### 4.3.2. Transgender (Trans Men/Trans Male Athletes) and Gender-Diverse Athletes on Testosterone Therapy

Less guidance exists amongst UK sporting organisations for the participation of trans men/trans male athletes on testosterone therapy, with many groups requiring either no information (hormonal or otherwise) or simply a written declaration. The Football Association is the only organisation requesting hormonal levels and requiring them to be within the male range. Exogenous androgens are known to be performance-enhancing for all athletes and are, therefore, on the WADA’s Prohibited List [58]. In hypogonadal male athletes, testosterone replacement therapy has often been found to result in testosterone levels above the upper range of normal [59], and its prevalence in mainstream sports has recently come under scrutiny for potentially conferring an advantage [60,61,62].

Beyond this, exogenous testosterone therapy may also confer an advantage within physiological levels [63,64,65], which requires further consideration. In males, testosterone levels fluctuate with the circadian rhythm and in response to disease, stress (both physical and mental), increasing age and other stimuli [66,67]. In some sports, rapid weight loss and low body fat levels are common prior to competitions (for example, in boxing) [68,69], with known effects on testosterone levels, as part of the recognised spectrum Relative Energy Deficiency in Sport (RED-S) [70,71,72,73]. Reductions in testosterone levels are observed when body fat is reduced below a certain level [74,75]. In trans male athletes receiving exogenous testosterone supplementation, such fluctuations or reductions pre-competition will not be observed, nor will diurnal fluctuations [76,77,78,79]. With dose–response effects from testosterone on strength, power, muscle size and body fat percentages within physiological ranges being demonstrable [57,65], athletes can titrate their levels to consistently remain at the upper limit of normal. Indeed, for this reason, the Nevada State Athletic Commissions (NSAC) banned testosterone replacement therapy in biological males in the Ultimate Fighting Championships (UFC) in 2014 [61,80].

### 4.4. Future Direction and Recommendations for TGD Sport Policies

The 2021 IOC Framework on Fairness, Inclusion and Non-discrimination promotes ten principles, primarily focusing on inclusion and non-discrimination, for sport federations to consider when organising sporting events [2]. Current considerations for the participation of transgender and gender-diverse athletes receiving gender-affirming hormone therapy (GAHT) in competitive sports are based predominantly on current serum testosterone levels. Testosterone-related performance enhancement may persist in some sports through the retention of physical advantage or muscular strength even after the normalisation of serum levels in trans women athletes [12,81]. The use of current testosterone levels has limitations, as previously discussed, and sport organisations should consider whether using current testosterone as a marker of competitive fairness fulfils their policy aims.

Further investigation into testosterone replacement in trans male athletes is required. This is of particular importance in weight-based sports, where the athlete might not be exposed to the testosterone fluctuations associated with weight changes that occur as part of the RED-S spectrum. The impact of anthropometric variables on sporting ability also likely requires further investigation. Clear policies for monitoring testosterone levels and anthropometric variables in TGD athletes should, therefore, be put in place by all sporting organisations if such policies enable TGD athletes to compete in a class other than that of their biological sex. A clear process for the management of levels lying outside of the proposed testosterone serum range should be developed, with clear guidelines on how abnormal results will be managed. Additionally, under the new 2021 IOC Framework on Fairness, Inclusion and Non-discrimination, male advantage over female athletes cannot be assumed in all sports. Studies have reported advantages in many sports in which power, strength and endurance are directly related to performance [25,26,27,28]. In some sports, further studies may be required; for example, motorsports or snooker. Sport organisations also need to specify the transitioning requirements for both pre- and post-puberty adolescent athletes, and all future transgender policies should include a pre- and post-puberty adolescent athlete section, as well as TGD policies for competitive and recreational sport. A section should also be included in all policies to define the term “transgender” for the purposes of the policy, and separate policies should be developed for intersex and DSD athletes. Additionally, when policies are finalised, consultation involving relevant stakeholders within the sport should take place and relevant external contributors should be allowed to offer input.

## 5. Limitations

This article provides a comprehensive overview of TGD athlete policies across UK sports and identifies areas of similarity and discrepancy. However, TGD policies outside the UK were not considered in this scoping review, although we recognise the recent TGD policy changes by the UCI [42] and Federation Internationale de Natation (FINA) [43], which will likely be reflected in UK TGD policies. Future scoping reviews could investigate and compare TGD policies from organisations outside of the UK. As different physiological requirements exist across various sports, TGD policies will likely differ between sports. It was beyond the scope of this paper to recommend TGD policies for each sport, and the paper rather aimed to highlight areas where a consistent approach to TGD athletes and sport participation is needed or could be useful. This review did not include policies related to para-athletes in UK sport; this is a topic for further investigation. TGD policy is a rapidly evolving field in sport medicine, and this scoping review therefore only provides a snapshot of sports’ current TGD policies, which are currently evolving.

## 6. Conclusions

This scoping review showed a lack of consistency and clear guidelines among UK sporting organisations in managing transgender and gender-diverse (TGD) athletes in competitive sports. Similar reviews investigating TGD policies outside of the UK should be considered. One area that policies for transgender athlete participation need to address is the appropriateness of using testosterone levels as a marker of performance advantage, as retained strength and anthropometric advantages are independent of current testosterone levels. Sporting federations need to develop their transgender policies with input from medical personnel, as well as the wider multi-disciplinary team and external agencies. Policies should consider the need for pre- and post-puberty guidelines, as well as guidelines for both competitive and recreational sport activities. Greater scrutiny needs to be given to trans men athletes regarding their testosterone levels, and sport-specific protocols should be established for how these testosterone levels will be monitored in transgender athletes, with clear rules for competitive sport exclusion based on them. Further studies on transitioning and transgender athletes aimed at understanding the impact of TGD athletes in their respective sports—and the potential impact of transitioning athletes on biologically female athletes’ safety, competitive opportunities and participation in sport at every level—will provide further information to help with policy making in this area. This is a challenging area for policymakers, and it is important that sport organisations are well-informed, undertake diverse considerations and consult widely in the further development and evolution of their sport’s TGD policy.

## Figures and Tables

**Table 1 ijerph-20-03568-t001:** The most popular sports (by participation numbers) in the UK and national governing bodies.

Sport	Sporting Organisation	Sport(s) Played within the Organisation	Male (M)/Female (F) Participation Split (%)	Policy Development
Football	The Football Association (FA) (England) Scottish Football Association (SFA) Football Association of Wales Irish Football Association (IFA)	Football	Proportion of population: 12.6% M, 1.2% F Relative participation: 90.8% M, 9.2% F	The Football Association Policy on Trans People in Football based upon: IOC 2004 statement Sports Council Equality Group “Guidance for National Governing Bodies of Sport” on “Transsexual People and Competitive Sport” Equality Act 2010 Gender Recognition Act 2004 FA Equality Manager and Medical Services team
Athletics	UK Athletics	Track and field sports; running (all types)	N/a	Policy based upon: World Athletics Eligibility Regulations for Transgender Athletes (2019) World Athletics Eligibility Regulations for the Female Classification (2019) Policy approved by UKA Board and owned by the Development Director
Cricket	England and Wales Cricket Board Cricket Scotland Cricket Ireland (Northern Cricket Union)	Cricket	N/a	Policy based upon: Advice from Stonewall UK IOC regulations ICC regulations Gender Recognition Act 2004 Overseen by ECB Head of Policy Development
Rowing	British Rowing	Rowing	N/a	Policy updated in 2022, based upon: NGB Policy IOC statement (2021)
Cycling	British Cycling	Cycling disciplines	Proportion of population: 14.4% M, 6.4% F Relative participation: 68.3% M, 31.7% F	Policy currently suspended/under review as of 06 April 2022, based upon: Equality Act 2010 Data Protection Act 2018 Gender Recognition Act 2004 IOC Consensus Meeting 2015 The UCI Regulations on Transgender Athlete Participation March 2020
Golf	England Golf Golf Ireland (including Northern Ireland) Scottish Golf Wales Golf	Golf	Proportion of population: 8.7% M, 1.3% F Relative participation: 86% M, 14% F	Policy based upon: Gender Recognition Act 2004 Equality Act 2010 Gender Recognition Panel, consisting of recognised gender specialists, as appointed by England Golf Human Rights Act 1998
Swimming	British Swimming	Swimming	Proportion of population: 13% M, 16.5% F Relative participation: 48.9% M, 51.1% F	Policy based upon: Sports Council Equality Group guidance British Swimming 2011 policy on transsexual competitors (previous policy)
Tennis	Lawn Tennis Association (LTA)	Tennis	Proportion of population: 3.2% M, 2% F Relative participation: 60.9% M, 39.1% F	Policy currently under review, based upon: Equality Act 2010 ITF Policy (2017) Policy approved by LTA’s CEO, executive team and board Authored by LTA Safe And Inclusive Tennis Team
Rugby union	Rugby Football Union (RFU) (England) Irish Rugby (IRFU) (Northern Ireland) Scottish Rugby Union (SRU) Welsh Rugby Union (WRU)	Rugby football (union rules)	N/a	Policy based upon: Equality Act 2010 Stonewall UK Authors acknowledged: Professor Dr Walter Pierre Bouman Professor John Arcelus
Rugby league	Rugby Football League (RFL) (England)	Rugby football (league rules)	N/a	N/a
Boxing	British Boxing Board of Control (BBBofC)	Boxing	N/a	No official policy
Snooker	World Professional Billiards and Snooker Association	Snooker Billiards	Proportion of population: 11.3% M, 3% F Relative participation: 77.9% M, 22.1% F	No official policy
Motor racing	Motorsport UK	All four-wheeled motorsports	N/a	No official policy
Darts	United Kingdom Darts Association	Darts	Proportion of population: 5.1% M, 1.5% F Relative participation: 76.8% M, 23.2% F	No information on policy development provided in official statement
Badminton	Badminton England Ulster Badminton (Northern Ireland) Badminton Scotland Badminton Wales	Badminton	Proportion of population: 3.4% M, 2.1% F Relative participation: 60.9% M, 39.1% F	Policy based upon: IOC Consensus 2015 Equality Act 2010 Data Protection Act 1998 General Data Protection Regulation 2018 Gender Recognition Act 2004 Approved by CMT and the board of directors Reviewed by ethics and compliance manager

**Table 2 ijerph-20-03568-t002:** UK sporting organizations’ transgender policies for the 15 most popular sports.

Sporting Organisation	Official Transgender Statement/Policy	Adolescent/Child Participation	Testosterone Cut-Off Levels	Evidence Required	Transgender Male and Gender-Diverse Athletes on Testosterone Therapy Competition Requirements	Transgender Female and Gender-Diverse Athletes on Oestrogen Therapy Competition Requirements	Years out of Competition (e.g., during Transition)	Notes
The Football Association	The Football Association Policy on Trans People in Football (2014)	No restrictions up until the age of 16 years	Normal range for affirmed gender as per the IOC	Individual case-by-case review Complete medical record Annual verification of hormones Proof of ID	Hormone therapy results in testosterone levels within biological male range	Hormone therapy/gonadectomy resulting in blood testosterone within the biological female range	Until such time as they have been provided with written clearance by the FA	Based on the International Olympic Committee’s position from May 2004. Without hormone therapy/surgery, cases are still reviewed individually
UK Athletics	UKA Eligibility for competition: Transgender and Female Classification Regulations (2021) UKA Policy and Procedures on Transsexual People in Athletics Competitions (2016)	Transgender female and gender-diverse athletes under 16, post-puberty: case-by-case review Transgender female and gender-diverse athletes, pre-pubescent: no restrictions	Normal range for affirmed gender, not specified	Serum hormonal levels Complete medical record Annual verification	Medical evidence that athlete is undergoing gender-affirming hormone therapy	Hormone therapy bringing serum testosterone within physiological biological female range or gonadectomy Can compete in male category sports if therapy has not begun	Not specified	Cases assessed on case-by-case basis following World Athletics guidance and may request support from World Athletics
England and Wales Cricket Board *	England and Wales Cricket Board Policy on Trans People Playing Cricket	Not specified	Not specified	Written clearance from the ECB Head of Policy Development	Cannot compete in any female-only competition, league or match	Written clearance from ECB Head of Policy Development	Not specified	Formed in line with guidance from the ICC, IOC and Stonewall UK (trans-rights charity). No restrictions in mixed sport
British Rowing	Trans and Non-Binary Inclusion Competition Policy and Procedures, 2022	“Due to the unique variance of physical and psychological developmental changes that take place during puberty and the medical options available to adolescents, the eligibility panel may, where appropriate, consider and approve an application without medical evidence… In considering an application the Panel will be mindful of the relevant criteria. It should be noted that a trans or non-binary person who has been determined eligible for the junior women’s category will, upon turning 16, be expected to comply with requirements set out [in the policy] and it is recognised that until the individual is able to access appropriate hormone treatment, they may not be eligible for the women’s category”	Not specified	Blood-measured testosterone levels	May compete as their affirmed gender in any “open” competition Need to be aware of anti-doping requirements	Must provide a signed letter stating that their consistent gender identity is “female”; this gender identity cannot be changed for a period of one year If >16 years old: testosterone level <5 nmol.L for previous 12 months and this level must be maintained during competition; medical evidence showing that the person has undergone surgery for the purpose of transitioning and therefore does not need to undergo hormone treatment; may compete as affirmed gender with evidence of serum testosterone or in male category competition if treatment has not started	Not specified. Testosterone levels for transgender females need to be <5 nmol.L for 12 months prior to competition	Adopted from the IOC position (2021). “A trans or non-binary person will no longer be eligible for the women’s category in domestic competition, if their testosterone levels rise above 5 nmol/l or if they have started female to male hormone treatment or undergone female to male gender reassignment surgery. Any trans or non-binary person who is not eligible to compete in the women’s category, will still be able to compete in the Open category.” “Submissions… must be made to an Expert Panel comprising two members of British Rowing’s Medical Advisory Panel, and a lawyer from one of British Rowing’s Panels.” Case-by-case decisions. “The Expert Panel may request that a trans or non-binary person whose gender identity was not female at birth, who is undergoing hormone treatment and is intending to compete in the women’s category, resubmit their declaration annually to confirm they are still undergoing treatment.” Eligibility for national teams and international competition differs from this domestic policy
British Cycling	British Cycling Transgender and Non-Binary Participation Policy (currently under review as of 6 April 2022, although data were taken from UCI guidelines updated in 2022) ^1^	Not specified	<2.5 nmol/L for at least 24 months	Recreational: no evidence required Competitive: race license to compete in proposed category; approval from medical professional	No evidence required other than a signed declaration of affirmed gender	Signed declaration and hormonal evidence, with serum testosterone remaining <2.5 nmol/L throughout competitive period	12 months minimum	IOC position taken into account, as well as UCI medical policy
England Golf *	Inclusion policy with respect to transgender people entering England Golf competitions	Not specified	Not specified	Proof of status >14 days in advance of competition Legal evidence +/− proof of hormone therapy	Evidence of legal recognition of re-assigned gender	Evidence of legal recognition of gender other than sex assigned at birth Hormone therapy for “sufficient” length of time	Female only: hormone therapy For sufficient length of time to minimise sex-related advantages	N/a
British Swimming	British Swimming Policy on Trans Competitors (under review) ^2^	<16, pre-pubescent: no restrictions <16, post-puberty: case-by-case review	Not specified	Evidence of pubertal stage (if < 16) Female only: hormonal levels	No hormonal or other evidence required	Hormonal therapy or verification of gonadectomy Annual re-verification of hormonal therapy	Female only: hormone therapy for sufficient length of time	Based on guidance from Sports Council Equality Group. All enquiries are presented to British Swimming’s Equality and Participation Panel. Separate guidance for contact disciplines
Lawn Tennis Association (LTA)	LTA policy and guidance on trans people playing tennis *	Not specified	Not specified Players on the performance pathway must be compliant with International Tennis Federation (ITF) policy	Proof of sex or Gender Recognition Certificate (GRC)	Not specified	Not specified	A GRC requires the person to have lived with the gender role for a minimum of 2 years	This document is currently under review in light of proposed Sport England guidance and equalities advice
Rugby Football Union (RFU) *	England Rugby Policy for the Participation of Transgender and Non-Binary Gender Players in Rugby Union (2019)	<11 age grade: no restrictions	<5 nmol/L	Written and signed gender declaration	Signed gender declaration	Signed gender declaration by the athlete Evidence of serum testosterone	**Transgender female and gender-diverse athletes:** testosterone <5 nmol/L for at least 12 months	No restrictions for non-contact (“tag”) rugby
Rugby Football League	RFL Transgender and Transsexual Policy	N/a	N/a	N/a	N/a	N/a	N/a	Not publicly available, published in November 2011 and referenced here: https://secure.rugby-league.com/ign_docs/Tackle%20IT%20Rugby%20League%20Factfile.pdf, accessed 1 August 2022
British Boxing Board of Control (BBBofC)	No official statement/policy	N/a	N/a	N/a	N/a	N/a	N/a	N/a
World Professional Billiards and Snooker Association	No official statement	N/a	N/a	N/a	N/a	N/a	N/a	World Women’s Snooker (WWS) Transgender Policy follows IOC guidance
Motorsport UK	No official statement	N/a	N/a	N/a	N/a	N/a	N/a	N/a
United Kingdom Darts Association	United Kingdom Darts Association Transgender Policy (recently removed from website)	N/a	Not required	Official government-issued ID that indicates their gender	Documentation that gender reassignment is ongoing	Documentation that gender reassignment is ongoing, 1 year minimum	**Transgender female and gender-diverse athletes**: one year since transition began	N/a
Badminton England *	Badminton England Policy for Transgender Badminton Players 2019 (reviewed 2021)	N/a	<10 nmol/L	Blood-measured testosterone levels, ongoing compliance	Can compete without restriction	Not specified	**Transgender female and gender-diverse athletes**: hormonal suppression for minimum of 12 months; non-compliance results in 12 month suspension	Based on IOC 2015, Gender Recognition Act 2004 and Data Protection 1998
International Olympic Committee (IOC)	IOC Statement on Transgender Guidelines (November 2015)	N/a	<10 nmol/L	Blood-measured testosterone levels	Can compete without restriction	Testosterone levels must remain below 10 nmol/L for the duration of time that eligibility to compete is desired Case-by-case approval Hyperandrogenism may permit participation in male events	Gender identity declaration cannot be changed for four years **Transgender female and gender-diverse athletes**: hormonal suppression for minimum of 12 months; non-compliance results in 12 month suspension	To require surgical anatomical changes as a pre-condition to participation is not necessary to preserve fair competition and may violate human rights

* Largest and most representative organisation in the United Kingdom. ^1^ As of 1 July 2022, international guidance from the UCI has altered their stance to reflect a lower maximum plasma testosterone of 2.5 nmol/L and extended their transition period for low testosterone from 12 to 24 months with respect to transgender female athletes [42]. ^2^ As of 19 June 2022, swimming’s world governing body, the FINA, have banned all female transgender athletes from international competitions if they are either >12 years of age or have experienced any part of male puberty beyond Tanner stage 2 [43].

## Data Availability

The data presented in this study are available on request from the corresponding author.

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
