# Peer review of "A Scoping Review of Transgender Policies in the 15 Most Commonly Played UK Professional Sports"

_ijerph, 2023, doi:10.3390/ijerph20043568_

Round 1

Reviewer 1 Report (Previous Reviewer 2)

No further comments

Reviewer 2 Report (Previous Reviewer 3)

Thank you for the opportunity to rereview this important paper. I have read it and the author careful and thoughtful responses to the reviewers, and I believe they are very satisfactory. I have only very minor editorial suggestions on the revision:

Line 58: “female’s”: this is an odd construction. I would say ‘women’s’ as a parallel to ‘trans women’ earlier in the sentence.

Line 90: Another odd sentence construction. I would say ‘as per the IOC, that is, “Someone…”'

Line 101: “had to be widely participated in”; consider ‘have high levels of participation’

Line 159: “transwomen” consider ‘trans women’ for consistency throughout the paper.

Line 156: “with the wording being very similar…”: consider: ‘with similar wording in policies across each sport’

These suggestions do not require re-review. I commend the author for their careful and important work.

This manuscript is a resubmission of an earlier submission. The following is a list of the peer review reports and author responses from that submission.

Round 1

Reviewer 1 Report

Thank you to the authors for their valuable contribution to this very current and pertinent discussion. I can see value in their contribution and summary around current sport policies, both from an applied by also future research perspective.  However, I have some significant concerns around some of discussion points and justification raised by the authors, which seem in part to go somewhat beyond the stated aims. This is particularly pertinent given the highly charged nature of these discussions and the implications for sport/activity and the TGD community moving forward.

With this context in mind please see comments below:

Definition of TGD: to ensure clarity within the piece and especially as it relates to discussion, and ensure readability for a broader audience a definition of TGD should be included.  Based on the discussion points raised is appears that you are considering Trans athletes, as opposed to Intersex/DSD or the broader consideration of 'gender diversity'.  Additionally, existence, consistency or lack thereof of definitions and inclusion/exclusions in the sport policies review should be noted.  This may be as simple as there is a lack of definition/consistency covered in the policies.

Line 66-69: A number of broad statements or examples are provided about 'known sex difference that relate to performance' however a number of the references provided to justify this point talk of difference, but are not necessarily linked to performance and/or athletes.

For example, Ferber et al = recreational runners, injury focused;  Jain et al = medical students; Lipps et al = attribute the different in reaction time to force thresholds; Schultz et al = nothing to do with athletic performance.

References should be updated or context clarified in the sense that there are 'differences' seen across population, not necessarily related to performance.

Line 68 Typo: thanthese 

Line 103: see first point above, unclear if this relates to trans AND gender diverse or just trans individuals - it appears so far that discussion specifically relates to trans and hormonal influences, as opposed to other gender diverse presentations (clarification warranted in the introduction to ensure clarity of understanding)

Line 127: see first point above, "Was the definition of TGD used by each sport considered?  If it varied, was that noted in review and what potential implications does that have for policies."

Line 228 - 245: The seems like a brief overview of a very complex area, which also doesn’t offer any counterpoint or discussion that there is also variation within the genders. References used again do not relate to athletes [41] which should be noted given the impact on training on performance and sex differences.

Line 254-255: Reference needed

Line 266-267: This is a very broad statement which is generalised across all sports - however proposed differences (and potential advantage) is not consistent across all sports and warrants noting here.

Line 278 - 279: This comments is missing context that is important for interpretation and the conclusions that the authors seem to be drawing.  For example - in Roberts et al., they also found that differences disappeared or were reduced. It was also specific to athletics (not all sports) and they note a number of limitations.

Line 281 - 282: The recommendation the authors attribute to Roberts et al, does not seem to be what Roberts et al. 2021 suggest. What they note is a difference 1-2 years, and that the 1 year currently mandated testosterone MAY be too short for athletics specifically.

Line 316-317: Reference missing

Line 329-330: Reference missing

Line 332: Reference missing

Line 337 - 338: This is poorly justified by the authors, and potentially damaging/misleading. Greater context is needed around the suggestion of issues for sporting safety e.g. there are variations in anthropometric characteristics within cisgender males and females - what makes it a safety concern when considering inclusion of TGD athletes? And references needed. 

Line 346: reference needed

Line 347-350: This concluding statement warrants furthers discussion and consideration from the authors - if they are advocating for policies for adolescent athletes then clearer justification needed and consideration of the potential harm of such policies should also be noted. As does the suggestion of policies for recreational sport.

Line 377: "the impact of transitioning athletes on the safety of biological female athletes"  There is the potential for this to be quite inflammatory and/or misrepresented. Little to no evidence or discussion has occurred in the manuscript that indicates any reason to be ‘concerned for safety of biological female athletes’ (or any consideration if there are potentially risks also to the transitioning athlete).  This should be rephrased or clarified.

Reviewer 2 Report

Dear Authors

I enjoyed reading your contribution. I only have a few comments.

p. 2 L68: Typo : replace 'thathheese' with ‘ these ‘

 p.13 L346: Please provide an example(s) of which sports may require further study.

Regards

Reviewer 1

Reviewer 3 Report

Thank you for the opportunity to review 'A Scoping Review of Transgender Policies in the 15 Most Commonly Played UK Professional Sports'. This is a well done, well-presented and much needed review on this important topic. The authors have done an excellent job in abstracting and clarifying existing policies and practices, and identifying existing gaps in respect of gender diverse athletics policies.The paper is well organised and clearly sets out the methodology and findings, both in text and in tabular form. The conclusions are supported by the findings, and there is clarifying discussion throughout. The Discussion included an excellent and well-organised analysis. The paper was a pleasure to read.

I note a couple of very minor editorial corrections ('thathheese') required in line 68 and formatting in line 442. I also hope the very useful tables can be formatted to minimise breaking across pages, because I'm sure these tables will be much copied and used. 

The one thing I would recommend and encourage the authors to consider is a brief sentence in section 5, the Limitations section, that says that this paper does not include policies related to gender diversity in para-athletes/athletes with disabilities. It appears that no organisations for disabled competitors were considered. One paper cannot do it all, but to overlook disabled competitors completely in 2022 is a significant omission itself. Simply acknowledge the omission (for now) and leave it for a future paper.  The other thing the authors might consider in the Conclusion section is a recommendation that similar scoping reviews be considered in other countries (but that may be stating the obvious).

Well done to all the authors.
